# A Study on Characteristic Emission Factors of Exhaust Gas from Diesel Locomotives

**DOI:** 10.3390/ijerph17113788

**Published:** 2020-05-27

**Authors:** Min-Kyeong Kim, Duckshin Park, Minjeong Kim, Jaeseok Heo, Sechan Park, Hwansoo Chong

**Affiliations:** 1Future Innovative R&D Strategy Department, Korea Railroad Research Institute (KRRI), Uiwang 16105, Korea; mkkim15@krri.re.kr; 2Transportation Environmental Research Team, Korea Railroad Research Institute (KRRI), Uiwang 16105, Korea; mjkim88@krri.re.kr (M.K.); jsheo1005@krri.re.kr (J.H.); sechani@krri.re.kr (S.P.); 3Department of Transportation System Engineering, University of Science & Technology (UST), Daejeon 34113, Korea; 4Transportation Environmental Research Institute, National Institute of Environmental Research, Incheon 22689, Korea; johnchong@korea.kr

**Keywords:** diesel locomotive, emission factor, EPA, emission reduction, portable emission measurement system (PEMS)

## Abstract

Use of diesel locomotives in transport is gradually decreasing due to electrification and the introduction of high-speed electric rail. However, in Korea, up to 30% of the transportation of passengers and cargo still uses diesel locomotives and diesel vehicles. Many studies have shown that exhaust gas from diesel locomotives poses a threat to human health. This study examined the characteristics of particulate matter (PM), nitrogen oxides (NOx), carbon monoxide (CO), and hydrocarbons in diesel locomotive engine exhaust. Emission concentrations were evaluated and compared with the existing regulations. In the case of PM and NOx, emission concentrations increased as engine output increased. High concentrations of CO were detected at engine start and acceleration, while hydrocarbons showed weakly increased concentrations regardless of engine power. Based on fuel consumption and engine power, the emission patterns of PM and gaseous substances observed in this study were slightly higher than the U.S. Environmental Protection Agency Tier standard and the Korean emission standard. Continuous monitoring and management of emissions from diesel locomotives are required to comply with emission standards. The findings of this study revealed that emission factors varied based on fuel consumption, engine power, and actual driving patterns. For the first time, a portable emission measurement system (PEMS), normally used to measure exhaust gas from diesel vehicles, was used to measure exhaust gas from diesel locomotives, and the data acquired were compared with previous results. This study is meaningful as the first example of measuring the exhaust gas concentration by connecting a PEMS to a diesel locomotive, and in the future, a study to measure driving characteristics and exhaust gas using a PEMS should be conducted.

## 1. Introduction

Diesel locomotive engines are safe and have a high combustion efficiency and fuel economy and a low flashpoint. They have been used extensively in Korea, but since the early 2000s their use has been gradually decreasing due to electrification and the introduction of high-speed electric rail. Currently, however, up to 30% of rail transport still uses 265 diesel locomotives and 83 diesel vehicles for carrying passengers and cargo [1,2]. The advantages of diesel locomotives include the traction of diesel engines and their ability to operate in mountainous terrain where power facilities are not easy to install. At present, diesel locomotive Series 7300 (GT26CW-2, Hyundai Rotem, Uiwang, Korea; EMD/GM, McCook, IL/Detroit, MI, USA), Series 7400 (GT26CW-2, Hyundai Rotem, Uiwang, Korea), Series 7500 (GT26CW, GT26CW-2, Hyundai Rotem, Uiwang, Korea; EMD/GM, McCook, IL/Detroit, MI, USA), and Series 7600 (PowerHaul PH37Acai, GE, New York, NY, USA) are in operation in Korea. Except for 7600 (PowerHaul PH37Acai, GE, New York, NY, USA), engines 16-645E3 (EMD, McCook, IL, USA) are used, mainly for towing freight trains. Diesel locomotives emit significant amounts of air pollutants, and research has demonstrated that air pollutants from diesel locomotives affect human health [1,2]. Primary railway-derived air pollutants include particulate matter (PM), nitrogen oxides (NOx), carbon monoxide (CO), and hydrocarbons (HCs) [3]. Railway transportation is a source of non-road emissions, and when diesel locomotive engine emissions are not controlled, the exhaust emissions can negatively affect communities and ecosystems. Considerable research has examined on-road automobiles and trucks, while air pollution from non-road mobile sources, such as diesel locomotives, has attracted relatively little interest. However, interest in technologies that reduce pollutant emissions from non-road vehicles is gradually increasing, as evidenced by the use of fuel activation devices (FADs) and portable emission measurement systems (PEMSs) [1,2,4,5]. Major countries such as those in Europe and the United States have also begun to regulate non-road vehicle emissions [1,2].

Diesel locomotives emit substantial amounts of fine PM (Dp < 2.5 μm). Although their contribution to the total amount of particle matter is small, the number concentration of particles is high. Research has demonstrated that fine PM affects human health [4,6,7,8,9]. Therefore, it is important to investigate the number concentration of fine particle sizes in diesel locomotive particulate emissions [10,11,12,13].

The present study analyzed the characteristics of exhaust gas from diesel locomotive engines. The emission factor of each exhaust gas was then compared with the United States Environmental Protection Agency (U.S. EPA) standard and the emission quality standard in Korea. This was the first time a PEMS was used at the National Institute of Environmental Research (2019) [2] to detect gaseous substances (NOx, CO, HCs) and PM for application to diesel locomotives. A PEMS is a device that can measure the emission gases in real time while simultaneously calculating the weight. The driving characteristics such as speed and acceleration and the emission gases can be simultaneously measured during operation. A PEMS is to utilize portable emissions measurement systems for on-vehicle emissions measurements. Additionally, a PEMS is a vehicle emission testing device that is small and light enough to be carried inside or moved with a motor vehicle that is being driven during testing, rather than on the stationary rollers of a dynamometer that only simulates real-world driving. This experiment is the first time that a PEMS has been used to measure the emissions of diesel locomotives in Korea. Measurement was carried out after 10 min, commencing with a warm start, with the engine power sequentially raised to notches 1, 5, and 8; the time required for measurement of the emissions was at least 5 min [1,2].

In this study, the emission factor was estimated based on the actual driving pattern of the Gyeongbu line (Seoul–Busan) in Korea. Based on this, it is significant in that the emission factors were compared with domestic and international standards.

## 2. Materials and Methods

### 2.1. Engine and Power

This experiment was used seven diesel locomotives (7331, 7352, 7356, 7418, 7437, 7443, 7457). Diesel locomotive Series 7300 (GT26CW-2, Hyundai Rotem, Uiwang, Korea; EMD/GM, McCook, IL/Detroit, MI, USA) and Series 7400 (GT26CW-2, Hyundai Rotem, Uiwang, Korea) were electric oversized diesel locomotive for passenger and cargo towing. The investigated Series 7300 and 7400 diesel locomotives are commonly used in Korea. A diesel locomotive consists of a diesel engine, a main generator, a traction motor, a bogie, and other attached structures. Table 1 lists the characteristics of the Series 7300 and 7400 diesel locomotive engines used in this study. The engines have a 2-cycle, 16-valve engine and adjustable throttle speeds (notches) from idle to notch 8. The throttle control has eight positions and an idle position. Each of the throttle positions is called a notch. Notch 1 has the slowest revolutions per minute (RPM), and notch 8 has the highest RPM. The revolutions per minute (RPM) of the diesel locomotive engine were 315 when idle, increasing to 900 RPM at notch 8. Diesel engine emissions are presented in terms of pollutants emitted at different brake powers instead of distance traveled [4,14], and the pulling capacity is shown in terms of brake horsepower per hour (bhph). The diesel locomotive measured 3127 mm × 20,787 mm × 4254 mm (W × D × H). Table 2 lists the diesel locomotive engine power and fuel consumption [15].

### 2.2. Experimental Methods

This experiment was carried out at the Busan Railway Vehicle Maintenance Team in charge of light and heavy maintenance of diesel locomotives in Korea. Diesel locomotive engine exhaust gases were measured according to SAE procedure J177 (Exhaust gas measurement procedure of diesel engine recommended by SAE International) and ISO 8178F mode (Exhaust gas measurement procedure of diesel engine in the non-road sector) as follows. The engine was ignited and warmed up for >10 min at a rated speed and load to stabilize it. The engine was then run in each experimental mode for >20 min at its steady state. The engine was stabilized so that the duration of constant discharge of the exhaust material was at least 10 min, and the time required for measurement of the emissions was at least 10 min. The CO, CO_2_, NO, NO_2_, and HC concentrations were monitored, and data collected in the last 3 min were averaged. The exhaust gas and PM characteristics were measured when idling and at notches 1–8, which corresponded to 0.6%, 5.6%, 14.4%, 28.1%, 35.8%, 48.6%, 61.7%, 81.6%, and 100% of the rated power, respectively (Table 2). The ISO 8178 F mode, a method for measuring non-road emissions and targeting locomotives and railcars, was used for measurement, taking into account the torque values of the existing SAE procedure J177 method (Table 3). Measurement was carried out after 10 min, commencing with a warm start, with the engine power sequentially raised to notches 1, 5, and 8. As shown in Table 3, ISO 8178 F mode should measure 5%, 50%, and 100% of the total load. Because the pull capacity is 3000, notches 1, 5, and 8 (corresponding to 150, 1500, and 3000 bhp) were measured in this experiment.

To investigate PM, the exhaust gas was passed through a filter and measured using gravimetric methods. NOx, a gaseous substance, was measured using an electrochemical method (EC), and CO was measured using a non-dispersive infrared method (NDIR). Total HCs were measured using a flame ionization detection method (FID). The engine power was repeated in the same order from idle to notches 1 to 8. Gaseous materials and PM were measured using a PEMS at the National Institute of Environmental Research (2019) [2]. After the pre-heating of the PEMS, ambient zero was performed and the flow meter was set to zero. Through the repeated tests using the PEMS, the trend was confirmed, and the reliability of the data was secured.

In this study, outliers were removed to select valid data. Outliers are values that are very small or very large compared to other data and are considered as outliers when they are smaller than “first quartile − 1.5 × quartile range” or larger than “third quartile + 1.5 × quartile range”.

Finally, the concentration of air pollutant emissions was measured, and emission factors were calculated while increasing the engine power of diesel locomotives from idle to notch 8. Furthermore, the emission factor was estimated based on the actual driving pattern of the Gyeongbu line (Seoul–Busan) in Korea. Later, the emission factors were compared with domestic and international standards.

### 2.3. Experimental Equipment and Measurement

Figure 1 presents the dilution and sampling devices used in this experiment. A secondary stack was installed in the diesel locomotive for purging, diffusing, and diluting the exhaust gas. The exhaust gas sampling tube was connected to the secondary stack. To prevent hot exhaust gas from condensing due to the temperature difference compared with the atmosphere, the sampling tube was insulated with a hot wire. The PEMS and gravimetric measuring equipment were both used to analyze PM. The filter used in the PEMS (Figure 2) was PTFE 47 mm, and the gravimetric method was used to measure the mass concentration with an 80 mm quartz microfiber filter. The filter weight was measured before and after by using an electronic balance, accurate to 0.0001 mg, and the measured mass value was divided by the total flow rate and converted into a mass concentration.

Isokinetic sampling of PM, discharged at a constant flow rate, was carried out by considering variables such as suction nozzle diameter, constant velocity suction coefficient, exhaust gas temperature, flow velocity, dynamic pressure, static pressure, orifice pressure difference, atmospheric pressure, and oxygen concentration. Isokinetic sampling is an equal or uniform sampling of particles and gases in motion within the stack. In particular, isokinetic sampling is very important for measuring PM.

The gaseous material was introduced through a sampling tube into a gas analyzer (Greenline 9000 gas analyzer, EUROTRON, Sesto San Giovanni, Italy) and measured as follows: NOx using an lectrochemical method, and CO using a non-dispersive infrared analysis method. Total HCs were measured using a gas analyzer (Toxic Vapor Analyzer 1000B, Thermo Electron Co., Waltham, MA, USA) and flame ionization detection was carried out at the load testing station of the Busan Vehicle Convergence Technology Group.

## 3. Results and Discussion

### 3.1. Exhaust Gas Concentration Analysis

The diesel locomotives examined in this study emitted PM averaging 43.4 ± 6.84 mg/m^3^ at idling and 239.0 ± 26.88 mg/m^3^ at notch 8. Figure 3a presents the concentrations of PM emitted at progressive notches. As the notch of the diesel engine increased, the concentration of PM emitted appeared to increase. Particulate matter is produced through pyrolysis, nucleation, surface growth, coalescence, and agglomeration of diesel fuel. As the engine notch increases, the amount of diesel fuel consumed increases. Accordingly, it was postulated that the production of the resulting PM would also increase [16].

The mass of PM was recorded using gravimetric measuring and averaged 4.87 ± 2.26 mg/m^3^ at notch 1 and 45.09 ± 2.26 mg/m^3^ at notch 8. The highest mass of PM was recorded at notch 6, after which PM emissions decreased with increasing engine power. This was judged to be due to the different engine characteristics of diesel locomotives compared with diesel vehicles. In the case of a diesel locomotive, the excess air ratio does not decrease when the output is increased, and after notch 6, the characteristics remain almost constant. Accordingly, it was concluded that the turbocharger operated at notch 6 supplied compressed air above atmospheric pressure, thereby reducing the extent of the decrease in excess air ratio [17].

As the amount of diesel fuel consumed increased as the engine notch increased, it was expected that the PM emissions would show the same trend. However, the emission concentration increased as the engine power rapidly increased from notch 6 to notch 8.

Nitrogen oxide concentrations are known to increase linearly as engine power increases and exhaust gas temperature rises [1,2,5]. Thermal NOx refers to nitrogen oxides generated by oxidation of nitrogen contained in combusted air at high temperatures [1,2,5]. As shown in Figure 3b, the diesel engine output was 147 ± 19.46 ppm at idle and 1137 ± 64.97 ppm at notch 8. Previous research has revealed that emissions of NOx can be reduced by slowing down fuel injection or by optimizing the timing of explosions [17]. The PEMS results from the National Institute of Environmental Research (2019) [2] exhibited the same trend, as the NOx concentration increased with increasing engine power from notch 1 to notch 8. NOx generated during diesel fuel combustion is mostly generated by thermal NOx mechanisms, and thermal NOx is more likely to be generated at higher temperatures. In summary, the emissions from diesel locomotives tend to increase when very high temperatures are applied, and emission concentrations increase as the throttle speed (notch) increases.

CO is formed when incomplete fuel combustion takes place due to a lack of oxygen. In general, CO is emitted during engine start-up and continuous acceleration. It can be caused especially at the time of starting and instantaneous acceleration of the engine where the rich mixtures are required. Diesel locomotives were found to emit high concentrations of CO at the idle state, which is the engine’s starting and initial operation, and during acceleration at notches 7 to 8 [18]. As shown in Figure 3c, CO concentrations were 94 ± 44.23 ppm with engine power at an idle state and 389 ± 86.60 ppm at notch 8. The PEMS results revealed that the CO concentrations were almost similar at notches 1 and 5, and the emission concentrations rapidly increased at notch 8.

HCs are composed of fuel that remains unburned due to insufficient temperatures near the engine cylinder wall. This is generally caused by an unbalanced ratio of air to fuel in the low load region [15]. In the case of diesel locomotives, it was confirmed that HCs were weakly increased regardless of engine power. As shown in Figure 3d, the emission concentrations were 4.3 ± 0.02 ppm with engine power at idle and 7.6 ± 0.13 ppm at notch 8. The PEMS results revealed that the engine power increased from notch 1 to notch 8, but increases in HC concentration were insignificant.

### 3.2. Calculation and Comparison of Emission Factors

This section explores the emission factors of diesel locomotives under various conditions of fuel consumption and engine power [19,20,21,22,23,24,25,26,27,28].

To calculate particulate and gaseous matter emission factors based on the fuel consumption of diesel locomotives, the emission concentration was multiplied by the discharge flow rate and divided by the fuel consumption volume. For gaseous matter, conversion factors of 46 g/22.4 L, 28 g/22.4 L, and 226 g/22.4 L were applied to the equations to convert the emission concentration in volume to weight. Additionally, a correction factor was applied to the equation to correct the discharge volume of the gaseous matter with temperature.

The emission factors of particulate and gaseous matter based on engine power were calculated using Equations (1)–(4) to compare the emissions of air pollutants from diesel locomotives with U.S. EPA Tier standards.
(1)$$\mathrm{PM}\left(\frac{g}{\mathrm{kw}\cdot h}\right)=\mathrm{emission}\;\mathrm{concentration}\;\times \;\mathrm{emission}\;\mathrm{flow}\times \frac{1}{\mathrm{engine}\;\mathrm{power}}$$
(2)$$\mathrm{NOx}\left(\frac{g}{\mathrm{kw}\cdot h}\right)=\mathrm{emission}\;\mathrm{concentration}\;\times \;\mathrm{emission}\;\mathrm{flow}\times \frac{1}{\mathrm{engine}\;\mathrm{power}}\times \frac{46}{22.4}\times \frac{273}{273+\mathrm{temperature}}$$
(3)$$\mathrm{CO}\left(\frac{g}{\mathrm{kw}\cdot h}\right)=\mathrm{emission}\;\mathrm{concentration}\;\times \;\mathrm{emission}\;\mathrm{flow}\times \frac{1}{\mathrm{engine}\;\mathrm{power}}\times \frac{28}{22.4}\times \frac{273}{273+\mathrm{temperature}}$$
(4)$$\mathrm{TC}\left(\frac{g}{\mathrm{kw}\cdot h}\right)=\mathrm{emission}\;\mathrm{concentration}\;\times \;\mathrm{emission}\;\mathrm{flow}\times \frac{1}{\mathrm{engine}\;\mathrm{power}}\times \frac{226}{22.4}\times \frac{273}{273+\mathrm{temperature}}$$

The U.S. EPA has established emission standards for air pollutants from light vehicles, heavy vehicles, and diesel locomotives [29,30,31,32].

In this study, the concentration of air pollutant emissions was measured, and emission factors were calculated, and the emission factors were compared with domestic and international standards.

For PM, the emission factor according to fuel consumption was 7.69 ± 2.19 g/L and the emission factor according to engine power was 2.29 ± 2.06 g/bhph. The emission factor for NOx according to fuel consumption was 49.54 ± 7.75 g/L and the emission factor according to engine power was 14.00 ± 10.44 g/bhph. The emission factor for CO according to fuel consumption was 5.09 ± 3.90 g/L and the emission factor according to engine power was 2.59 ± 5.15 g/bhph. The emission factor of HCs according to fuel consumption was 2.39 ± 1.24 g/L, and the emission factor according to engine power was 1.05 ± 1.84 g/bhph. Examination of the emission factors according to the engine power of particulate and gaseous matter revealed that they were higher than the U.S. EPA Tier 2 standard. In particular, the emission factor of NOx was high and will require continuous management and monitoring (Table 4).

In this study, the concentration of air pollutant emissions was measured, and emission factors were calculated while increasing the engine power of diesel locomotives from idle to notch 8. However, this method has limitations as it is not possible to evaluate the effects of changes in driving pattern and engine power during the actual operation of a diesel locomotive. Accordingly, the emission factor was estimated based on the actual driving pattern of the Gyeongbu line (Seoul–Busan) in Korea. The operation pattern of the Gyeongbu line (Seoul–Busan) was calculated as the ratio of the operating distance for each notch based on the KORAIL operation diagram and the total distance of the Gyeongbu line (Seoul–Busan). A diesel locomotive on the Gyeongbu line (Seoul–Busan) operates for 41% at idle power and about 25% at notch 8. The emission factor of diesel locomotives, taking into account the operation pattern of the Gyeongbu line (Seoul–Busan), is summed after multiplying the emission factors by notch and operating weight (Figure 4).

The total emission factor of PM based on the Gyeongbu line (Seoul–Busan) operation pattern was 4.27 g/bhph, which is significantly higher than the EPA Tier 2 standard of 0.13 g/bhph and the Korean emission standard of 0.2 g/bhph. The total emission factor of NOx based on the operation pattern of the Gyeongbu line (Seoul–Busan) was 23.58 g/bhph, which is significantly higher than the EPA Tier 2 and the Korean emission standard. The total emission factor of CO based on the operation pattern of the Gyeongbu line (Seoul–Busan) was 7.51 g/bhph, which is much higher than 2.0 g/bhph (EPA Tier 2) and 3.5 g/bhph (Korea). Based on the operation pattern of the Gyeongbu line (Seoul–Busan), the emission factor of HCs is 2.69 g/bhph compared with the EPA Tier 2 and Korean emission standard of 0.4 g/bhph.

## 4. Discussion

This study characterized diesel locomotive engine exhaust gases and determined concentrations and emission factors according to the notch (throttle speed) of the engine. In the case of PM, the emission concentration increased as the throttle speed increased. The emission factor of PM, based on both the driving pattern and the engine power, was found to be higher than the EPA Tier standard (0.13 g/bhph). NOx increased in concentration as temperature increased. Emission factors of NOx, which depended on the driving pattern and engine power, were about 3–7 times higher than the EPA Tier standard. CO was emitted at high concentrations at engine start and acceleration. The emission factor according to the driving pattern and engine power was slightly higher than the EPA Tier standard (2.0 g/bhph). Finally, the concentration of HCs increased weakly as the throttle speed increased, and the emission factor according to the driving pattern and the engine output was slightly higher than the EPA Tier standard (0.4 g/bhph).

In a previous study [33] of the off-road sector, the emission factors of diesel locomotives were compared with those of ships, and only the emission factors of PM and NOx were identified. Emission factors according to fuel consumption, engine output, and driving patterns were not examined.

## 5. Conclusions

In conclusion, the reduction of PM and gaseous emissions from diesel locomotives will require ongoing monitoring to minimize adverse effects on human health [34].

Altogether, this examination of fuel consumption, use of PEMS to assess gaseous materials and PM, and analysis of emission factors according to engine output, revealed that the EPA Tier standards were met or exceeded. This is the first time that a PEMS has been used to measure the emissions of diesel locomotives in Korea. A PEMS mainly measured the emission concentration in road vehicles, and this study has significance as the first study applied to non-road diesel locomotives.

Emission factors were also considered based on operating patterns and compared with domestic and international standards. More research is needed to examine emission factors by applying them to actual operating routes using PEMS that can simultaneously measure driving characteristics and emissions while driving.

## Figures and Tables

**Figure 1 ijerph-17-03788-f001:**
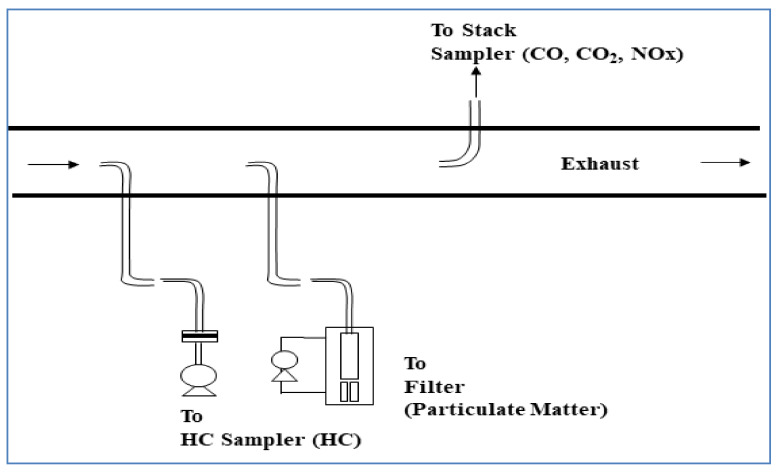
Schematic of the exhaust sampling and dilution system.

**Figure 2 ijerph-17-03788-f002:**
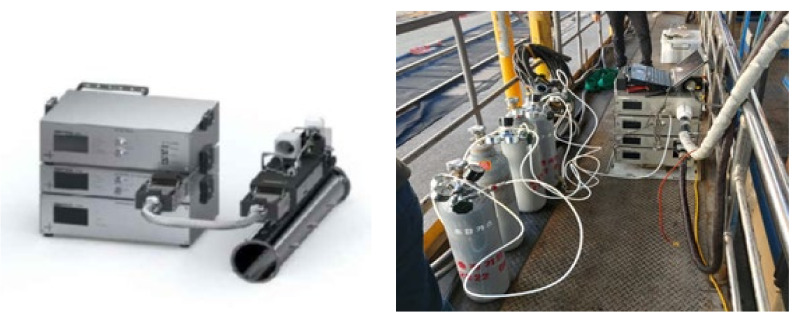
Photographs of the portable emission measurement system (PEMS).

**Figure 3 ijerph-17-03788-f003:**
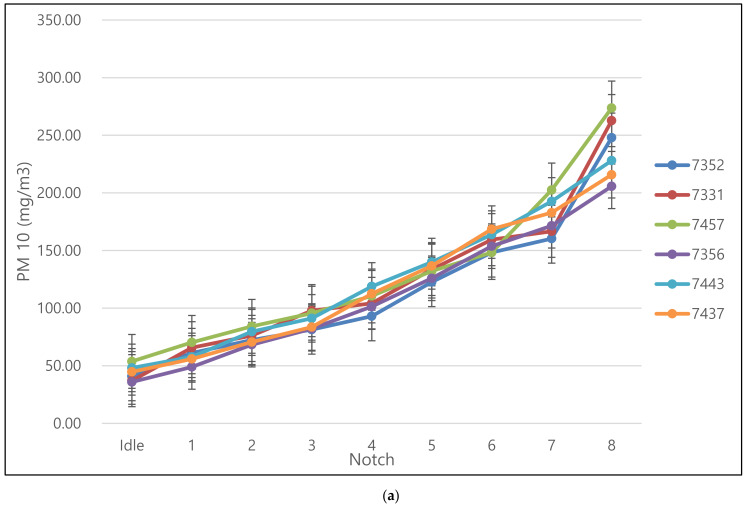
(**a**) Change in particulate matter concentration by notch; (**b**) change in nitrogen oxide concentration by notch; (**c**) change in carbon monoxide concentration by notch; and (**d**) change in total hydrocarbon concentration by notch.

**Figure 4 ijerph-17-03788-f004:**
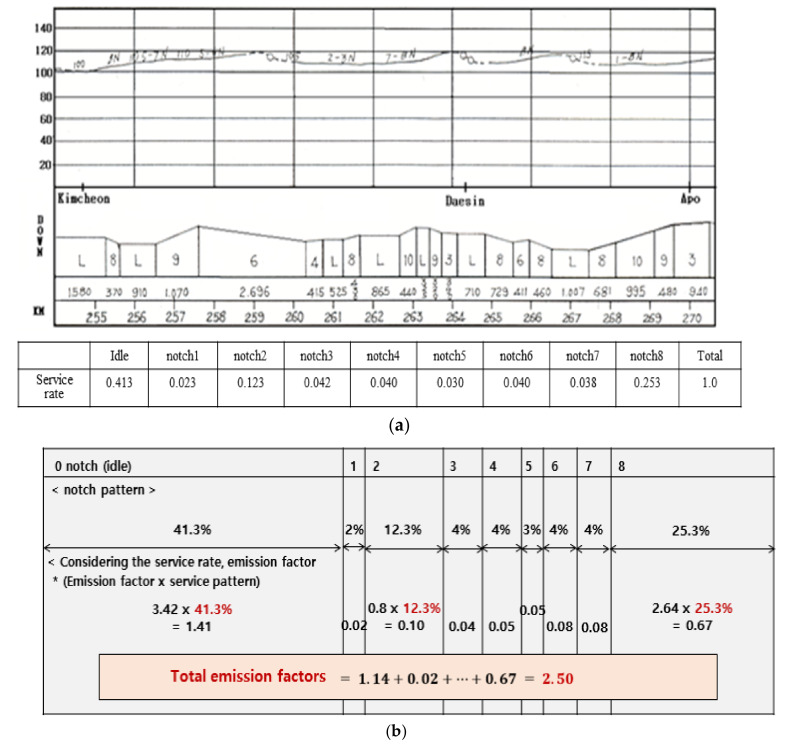
(**a**) The Gyeongbu line (Seoul–Busan) driving map and service rate by notch; (**b**) calculation of emission factors based on this operation pattern.

**Table 1 ijerph-17-03788-t001:** Characteristics of the Series 7300 and 7400 diesel locomotive engines used in this study.

Maker	Electric Motive Division, General Motor Company
Model no.	16-645E3
Cylinder size (mm)	230 × 254
Number of cycles	2
Compression ratio	14.5:1
Pull capacity (bhp)	3000
RPM idle/notch 8	(RPM idle) 315 (RPM notch 8) 900

RPM: revolutions per minute.

**Table 2 ijerph-17-03788-t002:** Characteristics of the diesel locomotive engine power and fuel consumption.

Notch	Power (bhp)	Rated Power (%)	Fuel Consumption (L/min)
Idle	17	0.6	0.4
1	173	5.6	0.6
2	426	14.4	1.4
3	830	28.1	2.6
4	1057	35.8	3.5
5	1434	48.6	4.8
6	1823	61.7	6.2
7	2409	81.6	8.4
8	2953	100	10.4

**Table 3 ijerph-17-03788-t003:** Non-road emission gas measurement method (ISO 8178 F mode).

Mode Number	1	2	3
Speed	Rated speed	Intermediate speed	Low–idle speed
Torque	100	50	5
Weighting factor	0.15	0.25	0.6

**Table 4 ijerph-17-03788-t004:** Emission allowance standards of major countries.

	U.S. EPA			Europe		Republic of Korea	
	Tier 2	Tier 3	Tier 4	Stage IIIa	Stage IIIb	Emission Standard	In This Study
PM	0.13	0.13	0.04	0.2	0.025	0.2	2.29 ± 2.06
NOx	7.38	7.38	1.74	7.4	4.0	7.4	14.00 ± 10.44
CO	2.0	2.0	2.0	3.5	3.5	3.5	2.59 ± 5.15
THC	0.4	0.4	0.19	0.4	4.0	0.4	1.05 ± 1.84

U.S. EPA: United States Environmental Protection Agency; PM: particulate matter; NOx: nitrogen oxides; CO: carbon monoxide.

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
