# Peer review of "A Study on Characteristic Emission Factors of Exhaust Gas from Diesel Locomotives"

_ijerph, 2020, doi:10.3390/ijerph17113788_

Round 1

Reviewer 1 Report

I recommend a publication of this study after revisions according to the following comments.

  1. The introduction of the paper should be further improved to highlight more valuable gaps compared to previous studies. At present, it is difficult to find something really novel in this introduction. Additionally, the expressions in the text, particularly the introduction, should be write more logically.
  2. The authors only detail their experiment design and some results, but it is suggested to highlight some findings different from previous studies and add more substantive discussion. Abstract and conclusions should be refined to show some more specific and valuable findings.
  3. In the section of “Materials and methods”, a too simple or incoherent description of the experiments makes it difficult to interpret how the data collected are validated and pre-processed. It is necessary to add some statistical parameters to understand the authors’ study results.
  4. Please rephrase the references that are suggested to be cited in a sequential order in the text, and the author should meet the requirement of journal format. Please recheck the whole text and ensure there are no obvious spelling and grammar mistakes, and present conventional expressions.

Reviewer 2 Report

The manuscript can not be accepted for publication in the current format, it should be restructured and written in a format suitable for the journal. Below I describe some of the aspects that should be improved, for possible publication in this journal.

  1. Only one locomotive was used?
  2. Describe the Field Study Design
  3. Describe some aspects of locomotives; for example passenger capacity, daily routes, type of engine etc.
  4. Describe the features and utilities of the Portable Emissions Measurement System
  5. A quality control section of the equipment is needed, how it was calibrated and how it was ensured that the data is representative of the actual expected concentration.
  6. Describe the Correction factors used to adjust for biases associated with the PEMS emissions measurement methods
  7. Describe how was Data Collection Procedure
  8. Include a Data Quality Assurance section
  9. Incluir Locomotive Emission Factors
  10. Line 145, authors say: “However, the emission concentration increased as the engine power increased from notch 6 to notch 8?”, how could be explained this behavior?
  11. I recommend including a graph that describes the behavior of each pollutant, for each notch
  12. Line 161: emit high concentrations of CO at the idle state, which is the engine’s starting and initial operation, and during acceleration at notches 7 to 8?, how could be explained this behavior
  13. Explain figure 4 well, what is it for ?
  14. The results must be discussed, under a more analytical view; that is, if the observed concentrations are within the characteristics of the fuels used, of the locomotive engine, of the design. In addition, these results must be compared with those observed in other similar studies inside and outside the studied site.

Reviewer 3 Report

The manuscript “A Study on Characteristics Emission Factors of Exhaust Gas from Diesel Locomotives” presents the assessment of emission of locomotive engines implemented in Korea, with evaluation of several pollutants, namely particulate matter, nitrogen oxides, carbon monoxide and hydrocarbons.

The aim of the study is clear and the topic of interest. However, organization of the manuscript must be improved before it can be considered for publication. Major comments are below:

Methods section is not complete and several details are described throughout the manuscript when corresponding results are mentioned. In particular, P7 about emission factors is totally missing in the Methods. In addition at P7, the meaning of Figure 4 is very unclear. According to the text, the table should present emission standards for air pollution according to the year of manufacture, but no indication of year is reported in the Figure. Please clarify all abbreviations used in the Figure (e.g. T- to T4, T1+, T0+).

Some details of pollutants are reported twice, e.g. P3L95-96 use of gravimetric methods and P5, L116-118.

In addition, only when presenting results in Figure 3, the reader realize that six locomotive engines are tested from the two different series 7300 and 7400. Thus, it is strongly recommended to revise Methods organization.

As regards study Results and Discussion, it would be more clear if sections are divided, with presentation of Results and then Discussion of study findings with comparison of literature data. Despite the content seems already presented, the organization makes very unclear to follow the different phases and results of the study.

In particular, data reported in Figure 3 are poorly reported since lines are very tight. A table would be a better presentation of these study results. In addition, at P5 L143-145, authors reported that PM emissions increased as the engine power increased from notch 6 to 8. However, based on Figure 3a, a linear increase can be noted from idle to notch 6, and then a higher increase seems present from notch 6 to 8. Please clarify the statement. In addition, when describing Figure 3d at P5 L12, authors reported that the increase in HCs is not significant. Although I do not recommend to use fixed cutpoints for statistical testing in line with the recent major trend involving the biomedical literature and the statistical epidemiologic methodology (see for example the 2016 American Statistical Association Statement, DOI: 10.1080/00031305.2016.1154108, and other recent papers PMIDs: 2898712, 27209009, 27272951, 28938715, 29650628 on the topic), it is unclear why authors reported an insignificant increase. The test implemented is not even mentioned. Figure 3d presents an evident though weak increase in emissions from idle to notch 8. Please check and revise the statement. Accordingly, please revise the related statement in Conclusions, P9 L259.

Results reported at P7 L207-213 are compared with data in Table 3. It would be more clear to add a column in Table 3 with emission factors in order to help the reader in understanding the comparison presented.

Minor suggestions:

References are not ordered (e.g. P1L38 ref [1, 35]). Please check and carefully revise.

Although interesting, Figures 1 and 2 might be reported as supplemental material.

Reviewer 4 Report

Minor changes to fix definitions
1: Describe notch (yes I can work it out but explain it to the readers)
2: PEMS (line 58) is used and only define much later (line 125)
3: Define isokinetic sampling (line 111)
4: Equations line 190-198 can use two equations. and add equation number numbers
5: Line 185, Yes I know M/22.4 but readers may not, similarly correction to 273K explain for the readers.

Must fixes
Need to state how long filters were sampled (line 60)
Is this really the first time diesel engines sampled globally/in Asia/Korea?

FIGURES need lots of attention
Fig 3, significant figures, what is legend on RHS?
Fig 4, I have no idea what you are showing me here. (Well I do but most of your readers won't)
Fig 5, unreadable, again I don't have time to work out what you are showing me and your caption doesn't tell me.

Round 2

Reviewer 1 Report

At present, it can be accepted for publication.

Reviewer 2 Report

The authors considered the recommendations

Reviewer 3 Report

Authors adequately addressed most of the comments during the revision. However, some minor issues are still present and need to be clarified.

Since seven locomotives are analyzed, why in figure results for three to six are  presented. Authors must clarify if data collection is missing for some of the locomotives. In addition, the use of the same colour throughout figure 3 for each locomotive would be helpful.

at P8, please clarify the matching between conversion factors and gaseous pollutants. I assume in the same order of equations, but please check and point it out. In addition, please harmonise abbreviations (i.e., HC and TC for hydrocarbons if I correctly understand.

Section 4 should be named "Discussion and conclusions"